# A Fast Algorithm for $k$-Memory Messaging Scheme Design in Dynamic Environments with Uncertainty

**Primary Keywords:** *None*

## Abstract

We study the problem of designing the optimal $k$-memory messaging scheme in a dynamic environment. Specifically, a sender, who can perfectly observe the state of a dynamic environment but cannot take actions, aims to persuade an uninformed, far-sighted receiver to take actions to maximize the long-term utility of the sender, by sending messages. We focus on $k$-memory messaging schemes, i.e., at each time step, the sender's messaging scheme depends on information from the previous $k$ steps. After receiving a message, the self-interested receiver derives a posterior belief and takes an action. The immediate reward of each player can be unaligned, thus the sender needs to ensure persuasiveness when designing the messaging scheme.

We first formulate this problem as a bi-linear program. Then we show that there exist infinitely many non-trivial persuasive messaging schemes for any problem instance. Moreover, we show that when the sender uses a $k$-memory messaging scheme, the optimal strategy for the receiver is also a $k$-memory strategy. We propose a fast heuristic algorithm for this problem and show that it can be extended to the setting where the sender has threat ability. We experimentally evaluate our algorithm, comparing it with the solution obtained by the Gurobi solver, in terms of performance and running time, in both settings. Extensive experimental results show that our algorithm outperforms the solution in terms of running time, yet achieves comparable performance.

## Introduction

The phenomenon of information asymmetry is commonly seen in many applications and has attracted extensive research attention from both computer science and economics. In these applications, an information sender can influence a receiver's behavior by strategically revealing information to them. Such interactions are usually modeled by the Bayesian persuasion framework (Kamenica and Gentzkow 2011). And in such environments, the information sender has an advantage in information, which often leads to an advantage in their reward or utility. For example, a navigation platform that has access to complete information about the traffic conditions of an area may recommend several routes to a user who only possesses local information. The user then chooses the best route based on the recommendations. The platform and the user may have misaligned goals, and the navigation platform can send route recommendations to influence the user's choice. Following the Bayesian persuasion framework, the platform can strategically design recommendation strategies to persuade users into taking specific actions that benefit the platform most.

Most existing studies only consider persuasions in a static environment. However, in real-world applications, the information sender and the receiver usually interact in a dynamic way. In this paper, we consider the persuasion model in a Markov decision process (MDP), where the sender has access to the state of the environment and the receiver is able to take action. We assume that both players are far-sighted and aim to optimize their accumulated rewards. The following example shows how the sender can improve their long-term reward by sending information to the receiver.

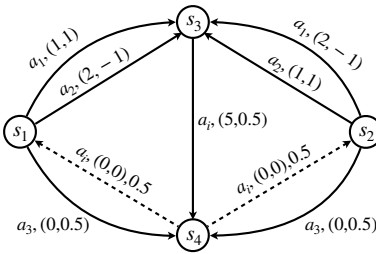

Figure 1: Rewards and state transitions for the MDP in Example 1

**Example 1.** *Consider the example MDP shown in Figure 1. The states $\{s_i\}_{i=1}^{4}$ are connected by directed edges indicating state transitions. Solid lines represent deterministic transitions and dashed ones probabilistic transitions. Each edge is labeled with the action triggering the transition and the immediate rewards for the sender and the receiver respectively. Dashed lines are also marked with the transition probabilities. There are 3 available actions $\{a_i\}_{i=1}^{3}$ for the receiver. Assume that the initial state distribution is 0.5 for $s_1$ and $s_2$, and 0 for both $s_3$ and $s_4$, i.e., the game will start at state $s_1$ or $s_2$ randomly. The discount factor for both players is 0.5. If the sender does not reveal any information to the receiver, the receiver will not be able to distinguish between $s_1$ and $s_2$, and thus will choose the "safe" action $a_3$ in the first step. The state then transits to $s_4$ deterministically. The*

*receiver always gets $0$ in state $s_4$ no matter which action the receiver chooses. And the state transits back to $s_1$, $s_2$ with equal probability. This process then repeats infinitely many times. As a result, the sender obtains reward 0.*

*However, if the sender reveals full information by telling the receiver what state the environment is currently in, the receiver will take action $a_1$ in state $s_1$ and $a_2$ in state $s_2$, leading to a strictly positive long-term reward for the sender. It is worth noting that the strategy of revealing full information is not the optimal one for the sender in this example.*

In this paper, we aim to design an information revealing strategy for the sender to maximize their long-term utility. In particular, we focus on the case where the sender uses a $k$-memory strategy, i.e., the strategy depends on the history of the previous $k$ steps. Since (Gan et al. 2022) already showed that finding the optimal Markov strategy in a similar setting is NP-hard, our main goal is to propose a fast algorithm that has a performance comparable to the optimal solution.

## Our Contributions

We formulate the problem as a bi-linear program and show that there exist infinitely many non-trivial persuasive messaging schemes for any problem instance. Moreover, we show that if the sender uses a $k$-memory messaging scheme, the optimal strategy for the receiver is also a $k$-memory strategy.

Then we propose an efficient heuristic algorithm based on backward induction and give a variant version when the sender has the ability to threaten the receiver. We conduct extensive experiments in both settings and the results show that our algorithm achieves a solution quality comparable to that of the solution found by the Gurobi solver, yet runs significantly faster.

## Related Works

Our paper is related to the broad area of information design, also known as "Bayesian persuasion". (Kamenica and Gentzkow 2011) study the setting where an informed sender aims to persuade an uninformed receiver in a static environment. This model has later been applied to many real-world applications, including security (Rabinovich et al. 2015; Xu et al. 2015), advertising (Badanidiyuru, Bhawalkar, and Xu 2018; Emek et al. 2014), and voting (Castiglioni, Celli, and Gatti 2020). More recently, this setting has been extended to a dynamic setting. (Farhadi and Teneketzis 2022; Ely 2017) consider a dynamic setting with a finite horizon where there are two states (one is absorbing), while we consider a more general environment with an infinite time horizon. (Celli, Coniglio, and Gatti 2020) consider a model where a sender interacts with multiple receivers in an extensive-form game. In their model, the sender reveals information to the receiver only once, while in our model, the sender sends messages to the receiver at every step. The most related paper is the study by (Gan et al. 2022), who capture the uncertainty in an environment with an external parameter. The key difference is that they focus on Markov signaling schemes, whereas we consider a more general $k$-memory messaging scheme. Actually, the Markov signaling scheme studied by

(Gan et al. 2022) is exactly equivalent to the 1-memory messaging scheme in our setting. They show that it is NP-hard to even approximate the optimal 1-memory messaging scheme against a far-sighted receiver. (Wu et al. 2022) design an efficient no-regret algorithm under an online learning setting. They aim to persuade a sequence of myopic receivers, while we consider persuading a single far-sighted receiver.

Our paper is also related to dynamic mechanism design (Papadimitriou et al. 2016; Pavan, Segal, and Toikka 2014; Athey and Segal 2007). In particular, recent work by (Zhang and Conitzer 2021) studies dynamic mechanism design in a finite horizon, where the mechanism designer, who has partial information about the state, aims to design a mechanism to elicit state information from an agent so as to make a better decision. On the contrary, we stand on the side of information design, studying how the sender can use this information advantage to maximize their utility. The common point is that we both adopt history-based strategies for the designer.

Another related topic is planning in MDPs. Particularly related to our work is (Zhang, Cheng, and Conitzer 2022), where the authors study a setting where an informed planner interacts with a self-interested agent with the choice to exit the environment. We both use history-based strategies. However, they impose participation constraints on the agent when the principle computes the optimal policy, while we need to guarantee persuasiveness constraints when the sender designs the optimal messaging scheme.

# General specifications

In the standard Markov decision process (MDP), a decision maker chooses an action at each time step to maximize their long-term reward. Now, consider a variant of MDP where there are two agents in the game, namely the *sender* and the *receiver*. The receiver can take action but has no access to the state. However, the sender can perfectly observe the state and send messages to inform the receiver about the state in order to influence their behavior. Both agents are rational and attempt to maximize their long-term expected utilities.

Formally, such a setting can be described by a tuple $\langle N, S, A, P, \rho_0, u, \gamma \rangle$, where:

- $N = \{s, r\}$ denotes the player set, where $s$ and $r$ denote the sender and the receiver, respectively.

- $S$ is a finite set of environment states, only observable for the sender.

- $A$ is a finite set of actions that the receiver can choose to take in each state. We assume all states share the same action set and let $d = |A|$ be the number of available actions.

- $P : S \times A \mapsto \Delta(S)$ is the state transition function. We use $P(s, a, s')$ to denote the probability that the receiver would arrival state $t'$ when he takes action $a$ in state $s$.

- $\rho_0$ denotes the initial state distribution, i.e., the initial state will be $s_i$ with probability $\rho_0(s_i)$.

- $u = (u_s, u_r)$, where $u_s : S \times A \mapsto \mathbb{R}_+$ and $u_r : S \times A \mapsto \mathbb{R}_+$ are the sender's and the receiver's immediate reward functions.

- $\gamma$ is a common discount factor.

We assume that the decision process repeats infinitely many time steps and consider the setting where the receiver can observe the immediate reward. Put differently, we assume that the receiver can speculate the state $s_t$ after taking action $a_t$, since the immediate reward $u_r(s_t, a_t)$ reveals information about $s_t$.[1] As a result, the receiver has a prior belief $\rho_{t+1} = P(s_t, a_t)$ about the next state $s_{t+1}$.

This setting induces a game between the sender and the receiver. The game proceeds as follows: the sender announces a messaging scheme at the beginning of the game, where a messaging scheme $(M, \pi)$ contains a message set $M$ and a policy $\pi$ specifying how a message is chosen. At each time step $t$, the sender first observes a state $s_t \in S$ and then sends a message $m_t \in M$ to the receiver according to the announced messaging scheme. Here, we assume that the sender has commitment power, i.e., the sender will never deviate from the announced scheme. After receiving the message, the receiver makes the best response to that message. Then the time step becomes $t+1$ and the state transits to the next one according to the transition function.

If two players are fully cooperative, i.e., their utilities align perfectly, then the sender can just send all the information they have, and the problem reduces to a standard MDP. However, the sender may only want to reveal partial information to the receiver, since the two players may have conflicting interests. We adopt the so-called Bayesian persuasion framework (Kamenica and Gentzkow 2011) to describe the sender's strategy.

### Histories and Messaging Schemes

The game between the two agents can be described by a game tree of infinite depth. The sender may use different messaging schemes at different tree nodes. In other words, the sender's messaging scheme can depend on the history information. We define $t$-length history $h = (s_1, a_1, \ldots, s_t, a_t)$ as a sequence of states and receiver's actions of the previous $t$ time steps. In this work, we mainly focus on the *k-memory* messaging scheme, which depends on the latest history with a length equal to or less than $k$. If $k = 0$, we call such a strategy a *Markov* strategy.

Denote by $\mathcal{H}_t$ the set of all histories of length $t$. Let $\mathcal{H} = \bigcup_{t=0}^{k} \mathcal{H}_t$ be the set of all histories with length no more than $k$, where $\mathcal{H}_0$ is the singleton containing the empty history $h_0$. At the beginning of the game, there is no history information but a prior distribution $\rho_0$ over the state set $S$. Thus the prior $\rho_0$ carries the same information as the empty history.

Given any $t$-length history $h$, we use $h + (s, a)$ to denote the new history by adding $(s, a)$ to the end of history $h$. Note that we may need to remove the earliest state and action to prevent the history length from exceeding $k$, i.e.,

$$h + (s, a) = \begin{cases} (s_1, a_1, \ldots, s_t, a_t, s, a), & \text{if } t < k \\ (s_2, a_2, \ldots, s_t, a_t, s, a), & \text{if } t = k \end{cases}.$$

A $k$-memory messaging scheme is a function that maps history-state pairs to distributions over the message space. Formally, denoted by $\pi : \mathcal{H} \times S \mapsto \Delta(M)$ the $k$-memory messaging scheme. We use $\pi(h, s, m)$ to denote the probability that message $m$ is sent by the sender when state $s$ is reached, given history $h$. Such a scheme is also called a "signaling scheme" in the literature (Kamenica and Gentzkow 2011).

Given history $h \in \mathcal{H}$, denote by $\rho_h$ the receiver's belief about the state $s$. As described in the previous section, $\rho_h(s)$ depends only on the state and action of the last time step, i.e., $\rho_h(s) = P(s_t, a_t, s)$.[2] We make the mild assumption that $\rho_h(s) > 0, \forall s$ throughout the paper. Once receiving message $m$, a rational receiver will derive a posterior belief over the state according to the standard Bayes rule:

$$\rho_h(s|m, h) = \frac{\rho_h(s) \cdot \pi(h, s, m)}{\sum_{s' \in S} \rho_h(s') \cdot \pi(h, s', m)}. \tag{1}$$

## Optimization Problem Formulation

We study how the sender can make use of this information advantage to influence the receiver's actions. The goal of the sender is to design a $k$-memory messaging scheme that maximizes their cumulative expected utility.

It is already known from (Gan et al. 2022) that solving the 1-memory messaging scheme design problem against a far-sighted receiver is NP-hard. Therefore, one cannot hope to find an efficient algorithm to solve this problem unless P=NP. In this section, we formulate the problem as a bilinear optimization problem which will be useful for later analysis.

In the above definition, we have no restriction on how many messages the sender can use. However, it is known that we can view each message as an action recommendation since each message induces a posterior belief of the receiver, which leads to a certain receiver action (Kamenica and Gentzkow 2011; Dughmi and Xu 2016). Thus the number of messages can be set equal to the number of actions without harming the sender's interest, i.e., $|M| = d$. In other words, given any messaging scheme, we can always construct an equivalent scheme $\pi$ with the message set $M_A = \{m_a : a \in A\}$, where each message $m_a$ corresponds to an action recommendation $a \in A$, achieving the same expected utility as the original messaging scheme.

**Persuasiveness.** Before giving a formal definition of persuasiveness, we first need to define the long-term utility for each player. Let $V_1^\pi(h, s)$ be the expected cumulative utility function when the sender uses strategy $\pi$ when the history is $h$ and the state is $s$. Similar to the Bellman equation (Bellman 1966), given a $k$-memory messaging scheme $\pi$, the cumulative expected utility function of the sender

---

[1]The receiver is able to perfectly identify $s_t$ in a non-degenerate case, i.e., $u_r(s_t, a_t) \neq u_r(s'_t, a_t), \forall s_t, s'_t \neq s_t, \forall a_t$.

[2]Assume that $h = (s_1, a_1, \ldots, s_t, a_t)$, then the previous state-action pair is $(s_t, a_t)$.

$V_1^\pi : \mathcal{H} \times S \mapsto \mathbb{R}$ should satisfy:

$$V_1^\pi(h, s) = \sum_{m_a \in M_A} \pi(h, s, m_a) \cdot \left[ u_s(s, a) + \right.$$
$$\left. \gamma \cdot \sum_{s' \in S} P(s, a, s') \cdot V_1^\pi(h + (s, a), s') \right]. \quad (2)$$

Given this, the overall expected utility of the sender from the beginning can be defined as follow:

$$V_1^\pi(h_0) = \sum_{s \in S} \rho_0(s) \cdot V_1^\pi(h_0, s). \quad (3)$$

Similarly, the receiver's long-term expected utility function $V_2^\pi : \mathcal{H} \times S \times A \mapsto \mathbb{R}$, under $k$-memory messaging scheme $\pi$ can be define as:

$$V_2^\pi(h, s, a) = u_r(s, a) + \gamma \sum_{s' \in S} P(s, a, s') \left[ \sum_{m_{a'} \in M_A} \right.$$
$$\left. \pi(h + (s, a), s', m_{a'}) \cdot V_2^\pi(h + (s, a), s', a') \right]. \quad (4)$$

Now we are ready to give a formal definition of persuasiveness:

**Definition 1** (Persuasiveness). *A $k$-memory messaging scheme $\pi$ is persuasive if it satisfies the following persuasive constraints, for all $h \in \mathcal{H}, m_a \in M_A, a' \in A$ :*

$$\sum_{s \in S} \rho_h(s) \cdot \pi(h, s, m_a) \cdot V_2^\pi(h, s, a)$$
$$\geq \sum_{s \in S} \rho_h(s) \cdot \pi(h, s, m_a) \cdot V_2^\pi(h, s, a'). \quad (5)$$

Simply put, a messaging scheme is persuasive if the receiver is always willing to take the recommended action, i.e., the recommended action always maximizes the receiver's long-term utility.

With the above analysis, we can now formulate the problem as the following mathematical program, with decision variables $\pi(h, s, m_a), V_1^\pi(h, s), V_2^\pi(h, s, a)$:

$$\begin{aligned}
\text{maximize} \quad & (3) \\
\text{subject to} \quad & (2), (4), (5) \\
& \sum_{m_a \in M_A} \pi(h, s, m_a) = 1, \forall h, s \quad (6) \\
& \pi(h, s, m_a) \geq 0, \forall h, s, m_a
\end{aligned}$$

Program (6) is a bi-linear program since constraint (5) is a bi-linear constraint.

## Theoretical Analyses

In this section, we analyze the problem in theory and derive some structural results. We first show that there exist infinitely many non-trivial persuasive messaging schemes for the sender, in any problem instance. Moreover, we show that the receiver can achieve optimality by using a $k$-memory

strategy if the sender also uses a $k$-memory messaging scheme.

In the standard Bayesian persuasion setting, there always exist trivial persuasive schemes, e.g., revealing full or no information to the receiver. Such trivial schemes also exist in our setting, but it is not clear if a non-trivial persuasive scheme exists, since our setting has much more complicated constraints. Before trying to find an optimal messaging scheme, we need to ensure that there indeed exist non-trivial persuasive schemes, since otherwise, there are only trivial schemes and we can just consider these special cases instead of searching the entire space.

To give some intuition about this result, we first consider a simple setting where $\gamma = 0$. We construct a trivial persuasive Markov messaging scheme as follows. Let $\beta_r^*$ be the optimal strategy of the receiver if they can observe the environment state $s$, i.e., $\beta_r^*(s) = \arg\max_{a \in A} u_r(s, a)$. We define the following Markov messaging scheme:

$$\pi^*(s, m_a) = \begin{cases} 1 & \text{if } a = \beta_r^*(s) \\ 0 & \text{otherwise} \end{cases}.$$

This messaging scheme is trivially persuasive since following the sender's recommendation already maximizes the receiver's utility. The proof of Lemma 1 is based on the above construction.

**Lemma 1.** *Assume that there are at least two actions $a_{i_1}$ and $a_{i_2}$, with corresponding states $s_{i_1}$ and $s_{i_2}$, such that $a_{i_1}$ and $a_{i_2}$ are the unique maximizers of $u_r(s_{i_1}, a)$ and $u_r(s_{i_2}, a)$, respectively. When $\gamma = 0$, there are infinitely many non-trivial Markov messaging schemes that are persuasive.*

The intuition behind the proof is that adding a small enough perturbation to a trivial scheme will not change the receiver's optimal strategy, thus maintaining persuasiveness. We defer the detailed proof into the appendix.

Then we show that infinitely many persuasive messaging schemes exist for any problem instance. Actually, this can be simply derived by applying the revelation principle (Myerson 1981) from the mechanism design literature. We also provide an alternative proof in the appendix that does not use the revelation principle.

**Theorem 1.** *For any problem instance, there are infinitely many persuasive messaging schemes.*

*Proof.* The intuition behind our proof is to "relabel" messages in any messaging scheme so that they correspond to the actual actions of the receiver. Let $(M, \pi)$ be any messaging scheme. If the sender uses this scheme, the receiver is then faced with an MDP as defined in the proof of Lemma 2. Let $\beta(h, m)$ be the receiver's optimal strategy in the MDP. Let $M_a(h) = \{m \mid \beta(h, m) = a\}$ be the set of messages that lead to the receiver's action $a$ when the history is $h$. According to the revelation principle, we can construct a new scheme that simply uses message set $M_A$ and replace each $m \in M_a(h)$ with $m_a$, and get the same receiver response $\beta(h, m) = \beta'(h, m_a), \forall m \in M_a(h)$. Thus the new scheme is persuasive. $\square$

In fact, the receiver is also faced with an MDP after the sender commits to a messaging scheme. Thus the problem studied in this paper is an MDP environment design problem for the sender. Based on this intuition, we have the following result.

**Theorem 2.** *When the sender uses a $k$-memory messaging scheme, the optimal strategy for the receiver is also a $k$-memory strategy.*

*Proof.* We prove this by showing that the receiver's problem can be viewed as an MDP. Since the sender has commitment power, their strategy will not change throughout the game. Thus the receiver can simply view the sender as part of the environment. From the receiver's point of view, they are faced with an MDP problem, where the environment of the MDP contains both the original environment and the sender. The state of the MDP contains both the history $h$ and the message $m$ sent by the sender.

After receiving a message $m$, the receiver will derive a posterior distribution by applying the Bayes rule:

$$\rho_h(s|h,m) = \frac{\rho_h(s)\pi(h,s,m)}{\sum_{s' \in S} \rho_h(s')\pi(h,s',m)}. \quad (7)$$

The expected immediate reward of the receiver for taking action $a$ is then $\sum_s \rho_h(s|h,m)u_r(s,a)$.

Formally, we can formulate the MDP faced by the receiver as follows:

- The state space is $S^* = \mathcal{H} \times M$;
- The actions spaces is $A^* = A$;
- The state transition function is $P^*((h,m), a, (h + (s,a), m')) = \rho_h(s) \cdot \sum_{s' \in S} \rho_{h+(s,a)}(s') \cdot \pi(h + (s,a), s', m')$;
- the reward function is $R^*((h,m), a) = \sum_{s \in S} \rho_h(s|h,m) \cdot u_r(s,a)$.

Since the sender uses a $k$-memory messaging scheme $\pi(h,s,m)$, the receiver's posterior belief of the environment state $\rho_h(s|m,h)$ only depends on the information of the previous $k$ steps. And even if the receiver uses a strategy that depends on a longer memory, they cannot obtain more information that can affect their behaviors. And in such an MDP, the receiver's optimal strategy is to choose an action for each MDP state $(h,m)$, which only contains information about previous $k$ time steps. □

## A Fast Algorithm for Finding $k$-Memory Schemes

In this section, we propose an efficient heuristic algorithm. The intuition behind our algorithm is as follows. The game proceeds in a Stackelberg way: the sender first announces their strategy and then the receiver follows. We view the game as a standard Bayesian extensive-form game as it provides a lower bound of the original game. However, the game still contains infinitely many steps. We further simplify the game by setting a parameter $T$ and only consider $T$ time steps. Thus the game tree has a maximum depth of $T$. We then modify the backward induction algorithm (Aumann 1995) and apply it to find a solution.

Backward induction is a strategy for analyzing a game by working backwards from the end to the beginning. The algorithm starts at time $T - 1$ and considers all possible $k$-length histories, of which there are $|\mathcal{H}_k|$ types of terminal nodes. Each node at this stage is labeled with the sender's messaging scheme, denoted as $\pi_h : S \times M_A \mapsto \mathbb{R}$. For each node, the optimal messaging scheme $\pi_h^*$ is computed, along with the expected utilities for both players. This information is then used to compute the optimal messaging scheme for the previous time period, time $T - 2$, and the process continues recursively until the optimal messaging scheme is determined for all nodes in the game tree.

Specifically, starting from time $t = T - 1$, we solve the following linear program for all nodes at time $t$, where each node can be uniquely identified by a history $h$:

maximize:
$$\sum_s \rho_h(s) \sum_{m_a} \pi_h(s, m_a)[u_s(s,a) + \gamma V_1(h + (s,a))]$$
subject to:
$$\sum_s \rho_h(s)\pi_h(s, m_a)[u_r(s,a) + \gamma V_2(h + (s,a))]$$
$$\geq \sum_s \rho_h(s)\pi_h(s, m_a)[u_r(s,a') + \gamma V_2(h + (s,a'))]$$
$$\forall m_a, \forall a',$$
$$\sum_{m_a} \pi_h(s, m_a) = 1 \qquad \forall s \in S,$$
$$\pi_h(s, m_a) \geq 0 \qquad \forall s \in S, m_a \in M_A. \quad (8)$$

Note that at any terminal node, there is no future reward thus we set $V(h + (s,a)) = 0$ at begin. At each backward step $t$, for each history $h$, after solving the above program, we obtain the optimal messaging scheme $\pi_h^*$ for node $h$. We let $V_1(h)$ equal to the objective of the program, and compute $V_2(h)$ as follow:

$$V_2(h) = \sum_s \rho_h(s) \sum_{m_a} \pi_h^*(s, m_a)[u_r(s, m_a) + \gamma V_2(h + (s,a))]. \quad (9)$$

In the end, we aggregate $\pi_h^*$ with all relevant histories $h$ and output a backward message scheme $\pi_{backward}$. Our detailed algorithm is listed in Algorithm 1.

### Threat Based Schemes

Our algorithm can also be applied to the setting where the sender is able to threaten the receiver. The receiver's utility is minimized when the sender provides no additional information about the underlying state, e.g., always sending the same message. If the sender threaten the receiver with a $k$-memory scheme, according to Theorem 2, such a threat lasts only for at most $k$ steps. In this section we consider threats that last forever.

When there is no information from the sender, the decision process of the receiver can be formulated as the following MDP $M^t = \langle S \times A, A, P^t, R^t \rangle$. In each step, the receiver only knows the prior belief about the environment state, which is actually the "state" in $M^t$. The transition

**Algorithm 1:** Finding a $k$-memory messaging scheme

**Input:** State set $S$, action set $A$, transition function $P$, initial state distribution $\rho_0$, reward functions $u_s$ and $u_r$, memory length $k$, discount factor $\gamma$.

**Parameter:** Backward step $T$.

**Output:** Message scheme $\pi_{backward}$.

1 Set $V(h + (s,a)) = 0$ for all terminal nodes $h$, and all $(s,a)$ state-action pairs;

2 **for** $t = T - 1, \cdots, 0$ **do**

3     **for** $h \in \mathcal{H}_k$ **do**

4        Solve the linear program (8) with existing $V(h + (s,a))$;

5        Save the message scheme $\pi_h^*$ and the expected utilities of both players;

6 Aggregate all $\pi_h^*$ to form $\pi_{backward}$;

7 return $\pi_{backward}$.

function $P^t$ is defined as follow:

$$P^t((s_{t-1}, a_{t-1}), a, (s_t, a_t)) = \begin{cases} \rho_h(s_t), & \text{if } a = a_t \\ 0, & \text{otherwise} \end{cases},$$

where $h$ is the history containing up to time step $t - 1$. Similarly, the reward function $R^t$ is defined as follows:

$$R^t((s_{t-1}, a_{t-1}), a) = \sum_s \rho_h(s) u_r(s, a).$$

Let $V^t(s, a)$ be the receiver's expected long-term utility starting from MDP state $(s, a)$. Following the standard approach (Manne 1960), we can find the solution to this MDP by solving the following linear program:

minimize:
$$\sum_{(s,a) \in S \times A} V^t(s, a)$$

subject to:
$$V^t(s, a) \geq \sum_{s'} \rho_h(s')[u_r(s', a') + \gamma \cdot V^t(s', a')]$$
$$\forall a' \in A, (s, a) \in S \times A.$$

The solution $V^t(s, a)$ to the above MDP is the best expected long-term utility the receiver can obtain when the sender does not provide any information. With such threat ability, the sender's persuasiveness constraints become:

$$\sum_{s \in S} \rho_h(s) \pi(h, s, m_a) V_2^\pi(h, s, a)$$
$$\geq \sum_{s \in S} \rho_h(s) \pi(h, s, m_a)[u_r(s, a') + \gamma V^t((s, a'))]. \quad (10)$$

We can thus find threat-based schemes for the sender by simply replacing the corresponding constraint in program (8) with the constraint (10) in Algorithm 1.

Note that the extra threatening ability does enlarge the sender's strategy space, as the $V^t$ is the lower bound of the receiver's utility. Replacing the original persuasiveness constraint with Equation (10) clearly makes the feasible region larger.

## Experiments

In this section, we experimentally evaluate our algorithm and report the experiment results. We compare our algorithm with the method of using Gurobi to solve the bilinear program defined in our paper, in terms of performance and running time. The experiment results demonstrate that our algorithm achieves solution quality comparable to that of the solution found by Gurobi, yet outperforms it in terms of running time.

We also conduct experiments with the sender being able to threaten the receiver. Due to space limitations, these results are deferred to the appendix.

**Experiment setup.** We conduct experiments on games with different sizes (number of states $\times$ number of actions), ranging from $2 \times 2$ to $12 \times 12$, and different discount factors $\gamma$, ranging from 0.1 to 0.9. Furthermore, we evaluate how the memory length influences the performance, by changing $k$ from 1 to 6. For each game size, we generate 20 game instances, where for each instance, the reward matrices of both players are generated randomly from the uniform distribution $U[0, 1]$, and the transition functions are also uniformly generated at random. All the algorithms are implemented with Python, and all the linear programs and bi-linear programs are solved using Gurobi (Python version, v9.5.2). All results with the same game size are based on the same set of reward matrices by varying $\gamma$ and $k$.

Since bi-linear programs are intractable to solve, we set the time limit parameter of Gurobi to 30 minutes (1800 seconds) when solving bi-linear programs, but do not limit the running time when solving linear programs.

We found that the Gurobi solver can hardly solve any bi-linear program of our generated game instances within the 30-minute time limit, even for $2 \times 2$ games. However, it can report the best feasible solutions obtained so far. Thus all the reported results in such cases are based on these feasible solutions.

All the results of our algorithm are obtained by setting the backward step to 100 ($T = 100$ in Algorithm 1) unless otherwise stated. Furthermore, all the reported results are averaged over the 20 randomly generated game instances.

**Performance.** We evaluate different algorithms' performance by comparing the expected utility of the sender obtained by them. We compare the performance of the two algorithms under different game sizes and different memory lengths. Since Gurobi does not even provide feasible solutions to the bi-linear program of some game instances in 30 minutes, the results are incomparable even if our algorithm can output feasible solutions. Thus all results are only average over the instances that Gurobi provides feasible solutions within 30 minutes. And we only compare the performance for games with sizes up to $5 \times 5$ and memory lengths up to 4, since Gurobi can hardly find a feasible solution for the bi-linear program of more complicated games.

Figure 2 shows the performances of two algorithms under different game sizes. Our algorithm achieves performances comparable to the bi-linear formulation. In general, for larger games, the sender can have higher utilities. Note that our algorithm sometimes achieves higher utilities than

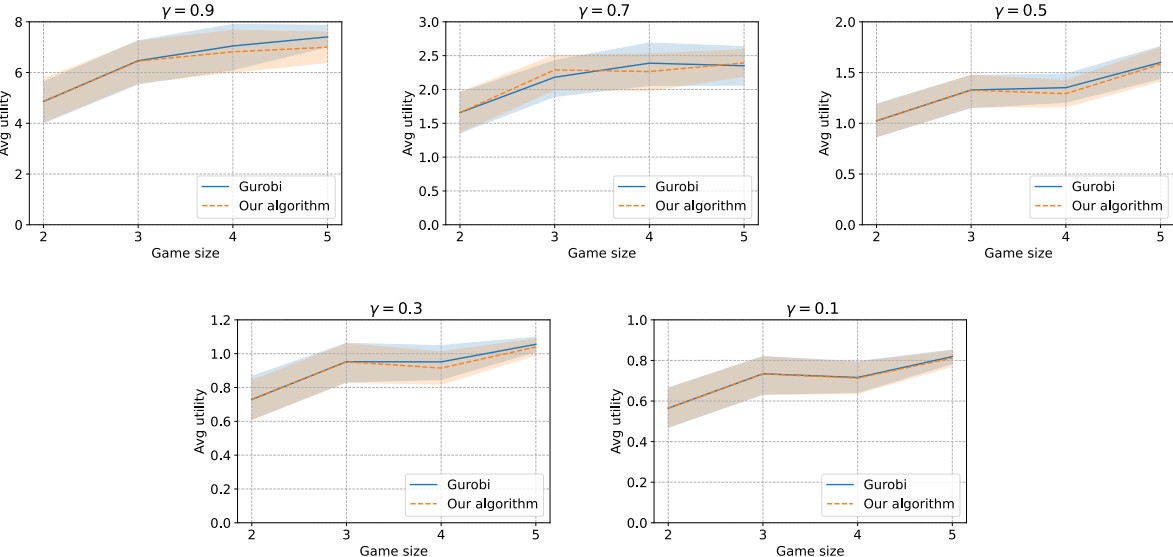

Figure 2: Average sender utility obtained by different algorithms with memory length $k = 1$.

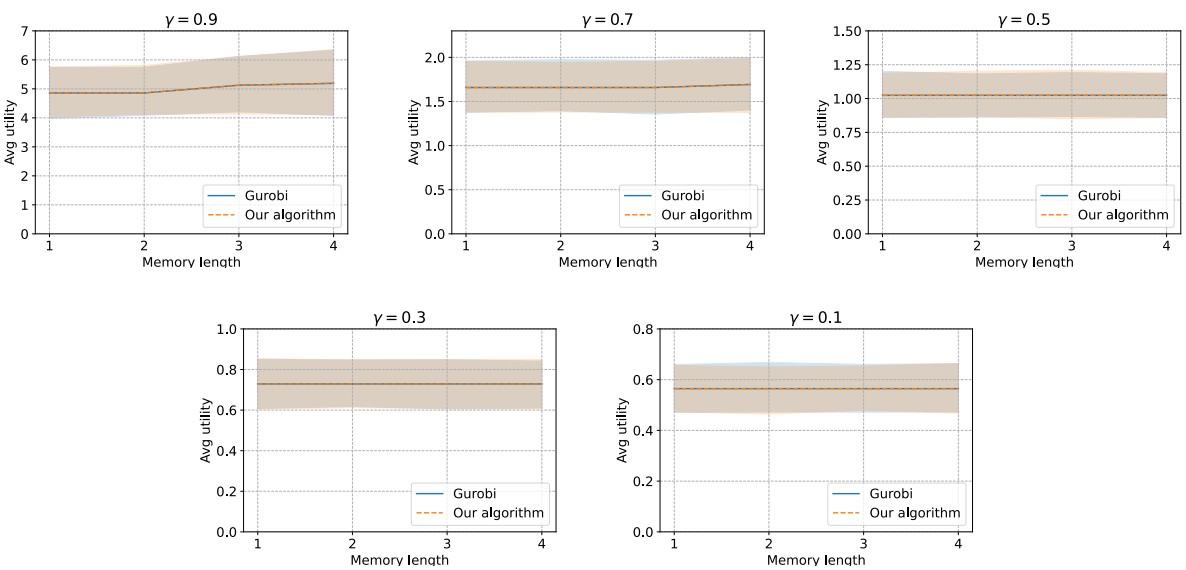

Figure 3: Average sender utility obtained by different algorithms in $2 \times 2$ games.

the bi-linear formulation simply because both algorithms only provide feasible solutions.

Figure 3 shows the performances of two algorithms with different memory lengths. The performances of the two algorithms are almost identical. When the discount factor is large, the sender can increase their utility by using a longer memory. But for small discount factors, the benefit of using a longer memory diminishes, as the receiver does not care too much about future utilities.

**Running time.** We analyze different algorithms' running times from three different aspects: (i) game size, (ii) memory length, and (iii) discount factor $\gamma$. Since Gurobi can hardly solve any bi-linear program in our experiments, we record how many of the 20 game instances that Gurobi can provide a feasible solution within 30 mins.

The results of solving bi-linear programs with Gurobi are shown in Table 1 and Table 2. It is clearly seen from Table 1 that as the game size increases, the number of games that Gurobi can provide a feasible solution decreases. Furthermore, this number also decreases when the discount factor $\gamma$ increases, which means that the more the receiver cares about long-term utilities, the harder it is for Gurobi to find a

Table 1: Number of games that Gurobi gives a feasible solution to the bi-linear program within 30 mins for $k = 1$.

| | Game size | | | | | |
| | 2 | 3 | 4 | 5 | 6 | 8 |
|---|---|---|---|---|---|---|
| 0.9 | 20 | 20 | 11 | 8 | 4 | 0 |
| 0.7 | 20 | 10 | 7 | 8 | 16 | 2 |
| $\gamma$  0.5 | 20 | 20 | 20 | 2 | 10 | 5 |
| 0.3 | 20 | 20 | 20 | 20 | 20 | 4 |
| 0.1 | 20 | 19 | 20 | 20 | 20 | 14 |

Table 2: Number of games that Gurobi gives a feasible solution to the bi-linear program within 30 mins for game size $2 \times 2$.

| | Memory length $k$ | | | | | |
| | 1 | 2 | 3 | 4 | 5 | 6 |
|---|---|---|---|---|---|---|
| 0.9 | 20 | 20 | 16 | 13 | 6 | 8 |
| 0.7 | 20 | 20 | 20 | 19 | 19 | 18 |
| $\gamma$  0.5 | 20 | 20 | 20 | 20 | 20 | 20 |
| 0.3 | 20 | 20 | 20 | 20 | 20 | 20 |
| 0.1 | 20 | 20 | 20 | 20 | 20 | 20 |

Table 3: Average running time (in seconds) of our algorithm for $k = 1$.

| | Game size | | | |
| | 2 | 3 | 4 | 5 |
|---|---|---|---|---|
| 0.9 | 0.542 | 2.604 | 9.120 | 25.449 |
| 0.7 | 0.536 | 2.600 | 9.048 | 25.381 |
| $\gamma$  0.5 | 0.528 | 2.562 | 8.945 | 24.981 |
| 0.3 | 0.531 | 2.560 | 8.885 | 24.819 |
| 0.1 | 0.531 | 2.553 | 8.891 | 24.829 |

Table 4: Average running time (in seconds) of our algorithm for game size $2 \times 2$.

| | Memory length $k$ | | | |
| | 1 | 2 | 3 | 4 |
|---|---|---|---|---|
| 0.9 | 0.532 | 2.122 | 8.439 | 33.237 |
| 0.7 | 0.537 | 2.126 | 8.486 | 33.364 |
| $\gamma$  0.5 | 0.529 | 2.087 | 8.335 | 32.795 |
| 0.3 | 0.529 | 2.115 | 8.361 | 33.097 |
| 0.1 | 0.526 | 2.088 | 8.354 | 32.819 |

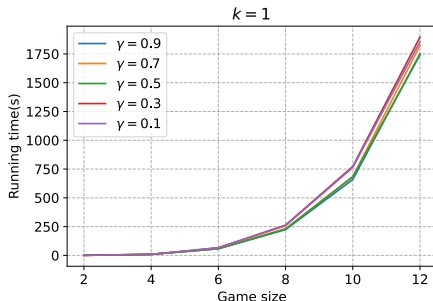

Figure 4: Average running time of our algorithm for $k = 1$ in games with different sizes.

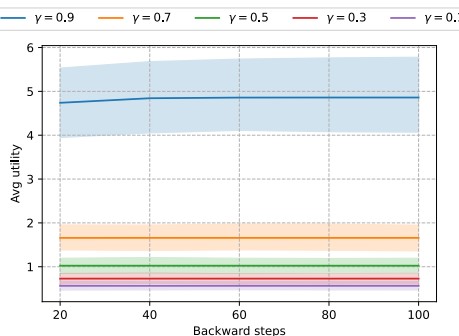

Figure 5: The average utility of our algorithm with $k = 1$, in $2 \times 2$ size games.

feasible solution.

As shown in Table 2, when the discount factor $\gamma$ is small enough, Gurobi is able to find feasible solutions for all the game instances with different memory lengths $k$. However, for larger discount factors $\gamma$, it becomes less likely for Gurobi to find a feasible solution within 30 mins as the memory length $k$ grows.

The results in Table 1 and 2 align well with our intuitions. As the game size and memory length increase, the strategy space of the sender grows larger. Therefore, solving these games becomes harder. Although the sender's scheme depends on previous time steps, it can also affect both agents' future utilities, since the receiver considers future utilities when making a decision and the current decision becomes past information in the future. With a larger $\gamma$, future utilities have a larger weight in the long-term utility and thus have more influence when the receiver chooses an action, making it difficult to find a good enough scheme.

We report the running time of our algorithm in Table 3 and 4. Our algorithm runs much faster compared with solving the bi-linear program. Our algorithm is able to find a feasible solution for all 20 game instances within 30 minutes, for all different game settings. In fact, our algorithm terminates within 30 seconds for most of the games.

We also conduct experiments to explore how large instances our algorithm can handle in 30 minutes, and record the corresponding average utility in different game sizes. Figure 4 shows that our algorithm can handle $12 \times 12$ games within 30 minutes. Unlike the bi-linear program formulation, the discount factor $\gamma$ actually has little impact on the running time of our algorithm. Changing the discount factor does not affect the execution of our algorithm except for the part of solving linear programs, which is also implemented using Gurobi. Thus we conjecture that the slight increase in running time is also due to the Gurobi solver.

**Hyperparameter.** We evaluate how the backward step affects the performance of our algorithms with $k = 1$, in instances with $2 \times 2$ game size and different discount factors. The results are provided in Figure 5. When $\gamma = 0.9$, the sender can obtain more utility by increasing the backward step from 20 to 40. Figure 5 also shows that increasing the backward step may not bring an obvious increase in utility, but may increase the running time quickly. Therefore, the backward step parameter can be used to balance the running time and the performance.

## Conclusion

We studied the problem of designing the optimal $k$-memory messaging scheme against a far-sighted receiver in a dynamic environment. We formulated this problem as a bi-linear program. Then we analyzed this problem in theory and derived some structural results. We also proposed a fast heuristic algorithm to solve this problem. Our experiment results show that the solution quality of our algorithm is comparable to that of the bi-linear program solved by Gurobi, and that our algorithm is much faster than solving the bi-linear program.

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
