# OpenReview forum: "A Fast Algorithm for k-Memory Messaging Scheme Design in Dynamic Environments with Uncertainty"
_icaps-conference.org/ICAPS/2024/Conference — ICAPS 2024_

### Official Review · Reviewer_M9uE · 2024-01-12

**Significance And Importance:** 2
**Soundness:** 3
**Novelty:** 3
**Clarity:** 3
**Overall Evaluation:** 1
**Confidence:** 3

**Weaknesses:**

1: Minor weaknesses that are easily fixable.

**Contributions Of The Paper:**

Post-rebuttal:

Thanks for the clarification. I would appreciate it if the discussion on the role of k was added to the paper. I also found your clarification wrt what threats mean more clarifying than what is in the paper -- if you have space, I recommend adding that to the final text too.

------------------------------------------------


The paper studies the problem of designing k-memory messaging schemes in
environments where a sender (who has a complete overview of the state space and
the transitions) can communicate with a receiver (who cannot observe anything
besides its current state, available actions, and received rewards). In this
problem, both the sender and receiver want to maximize their rewards. The
solution to this problem is a sequence of messages where the sender recommends
actions, and the actions are then taken by the receiver. We are particularly
interested in persuasive schemes: a scheme is persuasive if the receiver never
deviates from the action recommended by the sender. The authors show that the
problem of finding a persuasive scheme can be formulated as a bi-linear
program. They proceed to show that there are infinitely many non-trivial
persuasive k-messaging schemes (a messaging scheme is trivial if the sender
reveals either everything or nothing to the receiver). Moreover, whenever the
sender uses a k-memory scheme, the best strategy for the receiver is also
k-messaging.

Finally, the authors introduce an algorithm to compute k-memory schemes that do
not depend on bi-linear programming. The key insight of the algorithm is to
compute the scheme backward, from some time step T which now needs to be passed
as a parameter and solve a linear program for every step T-1, T-2, etc. In
practice, this algorithm scales better than Gurobi and, although heuristic, the
expected reward found by the algorithm is on par with the Gurobi solutions.

**Ethical Considerations:**

(1) Not Applicable: The paper does not have any ethical considerations to address

**Nomination For Best Paper:**

No

**Questions For Authors:**

- Why does k play no role in your experiments? Could you design benchmarks where
  it does indeed play a role? Would you have experiments showing it? If not, are
  1-messaging schemes sufficient in most cases? Would a greedy choice by the
  receiver be enough (without caring about previous messages) in most cases?

- Could you clarify my misunderstanding of the threat-based schemes? (See
  above.) Could you give a concrete example of its usefulness?

- Are you planning to make the source code public in case of acceptance?

**Reproducibility:**

0: N/A - nothing to reproduce.

**Strengths Of The Paper:**

The paper is well-written and the idea is quite interesting. The experimental
results give a clear picture of the usefulness of the tool (although I have some
remarks below). I am not an expert in the area, so I cannot fully evaluate the
novelty of the approach, but to the best of my knowledge, the key ideas are
indeed novel. The paper also fits the scope of ICAPS.

**Weaknesses Of The Paper:**

There are a few things that I did not fully grasp or found a bit unsatisfying in
the current version.

My main issue with the paper is that k does not influence all of your
benchmarks. One of the key insights of using the k-messaging scheme is to
exploit the last k steps. But in your benchmark, increasing k gives no advantage
whatsoever. It just seems from the experiments that the 1-message scheme is
enough in any case. I am not sure, but it seems to me like this is a flaw in the
benchmark design. My intuition says that one case where a larger k would be
useful in cases where the receiver is "stubborn": the sender keeps telling the
receiver to go from s1 to s2, but he keeps performing an action that self-loops
(with bad rewards, let's say) to s1, then the sender tries to move it to s3 to
see if it leaves from this bad state. But there is no parameter in your setup
where you can play with this. The generation of the randomized instances seemed
a bit too artificial to me.

Related to the last paragraph, the intro gives several applications and
scenarios where this sort of formulation could be useful. But then, in the
experimental results, all problems solved are randomly generated. While I do not
think this is crucial, I would have expected some domains closer to the things
you mentioned as motivation. At least, I think you should add one or two
sentences acknowledging that you did not do so and that there is no benchmark
for these problems (yet).

The part about the "threats" was the most unclear to me. Not sure I understood
what a threat would look like. If I understood it correctly, it is simply a
sequence of misleading messages so the sender reduces the reward of the
receiver. But why would this be the case? In which domains are threats
interesting or useful? What would such threats look like in practice? Perhaps
these questions are too basic, but it was not clear to me why the whole threat
bit is relevant.

Minor:

Everywhere: You use references as an "object" or a noun (e.g., "Since (Gan et
a. 2022) already..."). I would recommend rewriting it as "Since Gan et
al. (2022)" for example.

Optimization Problem Formulation: you say first that we want a scheme "maximizes
their cumulative expect utility." This is ambiguous: maximizes the sum of their
utilities or each wants to maximize their own? From the paper this becomes
obvious, but the first time it was misleading.

Equation (3): V^\pi_1(h_0) is overloading notation. V^pi_1 should have two
parameters by definition above. I believe you need a different symbol for the
left-hand side. (Not convinced the symbol is ever needed actually.)

After Equation (7): "action spaces" should be "action space"

After (8): "at begin" -> "at the beginning"

Performance: "we evaluate different algorithms" -> "we evaluate *the* different..." I believe.

References: Inconsistent naming of proceedings: for SODA and STOC you spelled
out the full name and added even the number of the proceedings -- this wasn't
done for AAAI for example.

References: almost half of the entries have missing page numbers. In particular,
all where the venue is written in short form.

Appendix: "defined above" -> "defined in the paper"?

---

> ### Author Rebuttal · Authors · 2024-01-27
>
> We thank the reviewer for the helpful comments. We will fix the typos in the next version.
>
> **Q1: Why does k play no role in your experiments?**
>
> Intuitively, the larger the parameter k, the higher the utility of the sender should be. However, based on the results of randomly sampled instances, we find that k has little effect on the sender's utility, especially when the discount factor is small. We also discuss (lines 531-537) this counter-intuitive discovery reflected in our experimental results.
>
> Here, we provide an example to show that using a k-memory with k>1 is useful. Although we do not prove this theoretically since the optimal solution is difficult to compute due to the hardness of the problem, we use Gurobi to find an upper bound for the example instance. The example is provided as follows.
>
> There are two states in the environment and two actions for the receiver. $P(s_1|s_i,a_i) = 0.3,  P(s_2|s_i, a_i) = 0.7, \forall s_i, a_i$.  $\rho(s_1) = 0.3, \rho(s_2)=0.7$. The payoff matrix is $u(s_1, a_1) = (1,1), u(s_1, a_2) = (0, 0), u(s_2, a_1) = (1, 0), u(s_2, a_2) = (0, 1)$. And the discount factor $\gamma = 0.5$.
>
> The upper bound of using 1-memory schemes obtained from Gurobi is 1.115. Now consider the following scheme: the sender uses scheme (1) at first and always uses this scheme when the receiver follows the recommendation. Once the receiver does not follow the recommendation, the sender turns into (2) forever.
> $$
> \pi(a_1|s_1) =1, \pi(a_1|s_2) = \frac{3}{7} - \epsilon \quad(1)
> $$
> $$
> \pi(a_1|s_1) =1, \pi(a_1|s_2) =1 \quad(2)
> $$
> The receiver's best response is always following the sender's recommendation. The sender's expected utility $U_s = \sum_{t=0}^{\infty} 0.5^t * (0.6 - 0.7 \epsilon)$. When $\epsilon $ approaches 0, $U_s = 1.2$, which is greater than the upper bound provided by Gurobi.
>
> **Q2: About the threat-based schemes**
>
> The sender has an information advantage about the state, and the receiver needs information provided by the sender to make better decisions. The most straightforward threat is that the sender does not provide any information (always sends the same message) to the receiver in the next k steps, or never if the memory length can be infinite. Intuitively, with the power of threat, the space of persuasive schemes is enlarged. In this case, the sender searches for an optimal solution in a larger space and may be able to get a better solution.
>
> **Q3: About the source code**
>
> Yes, we will release our code if accepted.

---

### Official Review · Reviewer_zCtw · 2024-01-21

**Significance And Importance:** 2
**Soundness:** 3
**Novelty:** 3
**Clarity:** 4
**Overall Evaluation:** 2
**Confidence:** 3

**Weaknesses:**

1: Minor weaknesses that are easily fixable.

**Contributions Of The Paper:**

The paper considers the problem of designing optimal k-memory persuasive messaging schemes when a sender aims to persuade a far-sighted receiver to take actions that maximize the expected utility of the sender. The setting is that the sender will commit to a pre-announced messaging scheme (message set and policy), and the goal is to find this optimal messaging scheme such that the receiver is always willing to take the recommended action. This is formulated in the paper as a bi-linear program. Then, the paper claims that there are infinitely many persuasive messaging schemes and thus requires searching the entire space of messaging schemes, and proves that the optimal strategy for the receiver is also a k-memory strategy if the sender uses a k-memory scheme. A fast backward induction algorithm is proposed to find a solution to the problem, as well as for a variant where the sender can threaten the receiver. Results show that this heuristic algorithm generate solution comparable to to solving a bi-linear program, and computes solution much faster than the bi-linear program.

**Ethical Considerations:**

(1) Not Applicable: The paper does not have any ethical considerations to address

**Nomination For Best Paper:**

No

**Questions For Authors:**

1. Clarification on the proof of Theorem 1 (detailed comment 1 and 2).
2. In what settings would the messaging scheme be given to the receiver a-priori with commitment?

**Reproducibility:**

3: Authors describe the implementation and domains in sufficient detail.

**Strengths Of The Paper:**

1. This is a well written paper that is easy to follow.
2. The paper looks generally theoretically sound (except for a question of Theorem 1, see below).
3. It is nice that the need to search over the space of messaging schemes is first shown before designing an algorithm to solve the problem. The proposed algorithm is reasonable and performs well compared to solving a bi-linear program, making it a good practical solution to the problem.
4. The experimental evaluation is well done and thorough.

The authors have addressed my concerns in the rebuttal. After reading the rebuttals and reviews, I am convinced the paper has a strong contribution. I have updated my score to an accept.

**Weaknesses Of The Paper:**

Detailed Comments:

1. The proof of Theorem 1 seems incomplete. It is clear from the proof that a new scheme constructed by replacing each m \in M_a(h) with m_a is persuasive, but it is not clear that this implies that there are infinitely many unique persuasive messaging schemes. It seems possible for some problem instance that all (M, \pi) are constructed into the same (or a finite set of ) messaging scheme with the technique in the proof as it is written.

2. Theorem 1 also doesn’t exactly show that there are infinitely many (non-trivial) persuasive schemes for a given message set M_a. Since the argument is that you can just use M_a as the message set, and M_a is the message set used for the algorithm, it seems important to prove it for the case of M_a.

3. The paper claims (lines 117-118) that it considers the general problem with infinite horizon, the algorithm actually only solves the T time step problem for a fixed parameter T.

Minor comments:

1. The condition/assumption of the sender announcing the messaging scheme at the beginning of the game and committing to it seems very strong. I understand that this is a common condition in the Bayesian Persuasion literature, but it is difficult to see where this sort of game can be applied.

2. It is not fully clear from the paper what the message space is, and I assume |M| < \infty?

3. V_1(h,s) should be defined as the expected cumulative utility function for the sender (line 278-280).

4. Equations 2-4 is a little ill defined, since V_1^\pi and V_2^\pi should depend on both the sender’s policy \pi_1 and receiver’s policy \pi_2, and not just \pi. a in equation 2 and a’ in equation 4 is not well defined without defining receiver’s policy \pi_2 (or receiver's optimal policy \pi*_2).

---

> ### Author Rebuttal · Authors · 2024-01-27
>
> We thank the reviewer for the helpful comments.
>
> **Q1: Clarification on the proof of Theorem 1.**
>
> Thanks for the valuable comment. A message scheme consists of a message set and a policy $\pi$. It is possible that the constructed message set $M_a(h)$ is the same, but the probability of sending message $m_a$ could be different for different $\pi$. Since there are infinitely many $\pi$ functions, the number of constructed schemes is also infinite.
>
> For the second concern about Theorem 1, the message set $M_a$ is also designed by the sender instead of given. According to the revelation principle, each message can be regarded as an action recommendation. Thus there is a one-to-one correspondence between the message set and the action set, i.e., $|M| = |A|$ (lines 268-270).
>
> **Q2: In what settings would the messaging scheme be given to the receiver a-priori with commitment?**
>
> This assumption is standard and common in game theory, particularly in Stackelberg games. It is often the case that the messaging scheme is announced to the receiver reactively rather than proactively. For example, in a recommender system, a user does not observe the recommendation algorithm beforehand. Instead, the user empirically estimates the behavior of the algorithm through interactions with the platform. In such a case, the commitment power of the sender is not a ``power'', but a kind of constraint that the sender must consider when designing messaging schemes.
>
> We also answer some other concerns mentioned by this reviewer.
>
> - $V_1(h, s)$ should be the cumulative utility function for the sender.
>
> Yes. We will make it clear in the next version.
>
> - Equations 2-4 are a little ill-defined since $V_1$ and $V_2$ should depend on both the sender's policy and the receiver's policy.
>
> Given a persuasive scheme $\pi$, the receiver's optimal action is $a$ upon receiving message $m_a$. Therefore, within the space of persuasive message schemes, both functions $V_1$ and $V_2$ are well-defined if a persuasive scheme $\pi$ is given.
>
> - The paper considers the infinite horizon setting, but the algorithm only solves the T time step problem for a fixed parameter T.
>
> The reason is twofold. First, the infinite horizon problem is impossible to solve and we have to approximate the solution using only a finite horizon. Second, because of the discount factor, when $T$ is large enough, the reward after $T$ has very little effect on both the receiver's utility and behavior.

---

### Official Review · Reviewer_YwbY · 2024-01-22

**Significance And Importance:** 2
**Soundness:** 4
**Novelty:** 3
**Clarity:** 4
**Overall Evaluation:** 2
**Confidence:** 5

**Weaknesses:**

0: Minor weaknesses requiring some work to be addressed for the paper to be accepted.

**Contributions Of The Paper:**

The authors study a variant of the Bayesian persuasion (which has 1000+ variants already) in sequential decision making, where the sender’s messaging scheme depends on information from the k most recent steps. The model follows that of Gan et al. (Bayesian Persuasion in Sequential Decision-Making, AAAI 2022).

The contributions are multi-fold:
-- There are some minor structural results proven.
-- There is a discussion of computational complexity, albeit not very convincing. It is well known that there are cases of bilevel and multi-level optimization that are easily solvable (e.g. https://arxiv.org/abs/1702.03999, https://arxiv.org/abs/2105.09407), and for a particular bi-level optimization problem, the authors should show that it is still hard.
-- An efficient heuristic algorithm is presented. The heuristic considers a standard Bayesian extensive-form game as a lower bound of the original game, and limits the maximum depth of the game tree to a constant. Then, they apply the backward induction algorithm (Aumann 1995).

**Ethical Considerations:**

(5) Excellent: The paper comprehensively addresses all of the applicable ethical considerations

**Nomination For Best Paper:**

No

**Questions For Authors:**

Would the code be accessible in pastebin or similar?

**Reproducibility:**

2: Some details are missing, but the paper still appears to be replicable with some effort.

**Strengths Of The Paper:**

The structural results are nice, if minor, and technically correct.

The heuristic itself is rather straightforward, but also easy to understand.

Having read the rebuttal, I am confident that the authors should be able to prove the complexity results for the final version.

**Weaknesses Of The Paper:**

The discussion of the complexity of the bi-level optimization problems is not particularly clear. The authors should explain the model of Gan et al and the changes they made and why they think it does not impact the complexity.

The discussion of the use of Gurobi is muddled. The authors should really refer to the results as to the results of their heuristic. "Gurobi" is often associated with exact methods, and indeed, the bi-level optimization problem could be formulated as a MIP and solved with Gurobi exactly. This is not explained, though.

There is no code provided, so it is impossible to check the experimental results.

---

> ### Author Rebuttal · Authors · 2024-01-27
>
> We thank the reviewer for acknowledging our contributions.
>
> **Q1: The discussion of the complexity of the bi-level optimization problems is not particularly clear and the authors should explain the model of Gan et al.**
>
> Thanks for the valuable comment. The hardness result is NOT because we have a bi-linear formulation. Instead, the complexity of this problem stems from the fact that the setting considered by Gan et al. is a special case of ours.
>
> Both their paper and this paper consider the setting with uncertainty in MDPs. In their model, the receiver observes the state $s$. The uncertainty comes from an external parameter $\theta$. In our model, the uncertainty directly comes from the state $s$ and the receiver observes the last state-action pair. Therefore, the observable state $s$ and the unobservable parameter $\theta$ in their model can be regarded as the state-action pair and the current state in our model. Viewing from this angle, their model is essentially the same as ours with memory length k=1.
>
> We will certainly make this clear in the next version, along with a formal discussion of the relationship between these two models.
>
> **Q2: The discussion of the use of Gurobi is muddled.**
>
> Thanks for pointing this out. The results labeled ``Gurobi'' mean that we directly solve the bi-linear program (Equation 6) using Gurobi. In our heuristic algorithm, we also use Gurobi to solve a linear program (Equation 8).
>
> We will clarify this in the next version.
>
> **Q3: Would the code be accessible in Pastebin or similar?**
>
> We will certainly release our code upon acceptance.

---

### Meta-Review · Area_Chair_SNQW · 2024-02-01

**Recommendation:** Accept (Oral)
**Confidence:** 5

**Metareview:**

This paper looks at a Bayesian persuasion problem, where the receiver plans for the long term and the sender uses a k-memory messaging scheme. Although Bayesian persuasion is usually studied in game theoretic settings, this paper looks at the long-term problem, and presents a bi-linear optimization approach which can compute an optimal policy, making this also suitable to ICAPS.

When preparing the final version of the paper, please take the reviewers' comments into account - I am confident they will improve the paper even more.

**Ethical Considerations:**

(1) Not Applicable: The paper does not have any ethical considerations to address